# Scalable Distributed State Estimation in UTM Context

**DOI:** 10.3390/s20092682

**Published:** 2020-05-08

**Authors:** Marco Cicala, Egidio D’Amato, Immacolata Notaro, Massimiliano Mattei

**Affiliations:** 1On-board Systems and ATM Department, Italian Aerospace Research Centre CIRA, 81043 Capua (CE), Italy; 2Department of Science and Technology, University of Naples “Parthenope”, 80143 Napoli, Italy; egidio.damato@uniparthenope.it; 3Department of Engineering, University of Campania “L.Vanvitelli”, 81031 Aversa (CE), Italy; immacolata.notaro@unicampania.it (I.N.); massimiliano.mattei@unicampania.it (M.M.)

**Keywords:** UAS traffic management, multiple UAV navigation, navigation in GPS/GNSS-denied environments, distributed state estimation, consensus theory

## Abstract

This article proposes a novel approach to the Distributed State Estimation (DSE) problem for a set of co-operating UAVs equipped with heterogeneous on board sensors capable of exploiting certain characteristics typical of the UAS Traffic Management (UTM) context, such as high traffic density and the presence of limited range, Vehicle-to-Vehicle communication devices. The proposed algorithm is based on a scalable decentralized Kalman Filter derived from the Internodal Transformation Theory enhanced on the basis of the Consensus Theory. The general benefit of the proposed algorithm consists of, on the one hand, reducing the estimation problem to smaller local sub-problems, through a self-organization process of the local estimating nodes in response to the time varying communication topology; and on the other hand, of exploiting measures carried out nearby in order to improve the accuracy of the local estimates. In the UTM context, this enables each vehicle to estimate both its own position and velocity, as well as those of the neighboring vehicles, using both on board measurements and information transmitted by neighboring vehicles. A numerical simulation in a simplified UTM scenario is presented, in order to illustrate the salient aspects of the proposed algorithm.

## 1. Introduction

Over the last few years, Small Unmanned Aircraft Systems (sUAS) have experienced a widespread diffusion both in military and civilian applications. Their diffusion is destined to grow even further, since they are capable of operating close to the ground and overcoming obstacles in all sorts of hazardous conditions forbidden to traditionally manned vehicles. sUAS offer new opportunities in different operational scenarios including public safety, search and rescue, disaster relief, infrastructure monitoring, precision farming and delivery of goods [1]. Most sUAV operations would take place in low-altitude, densely occupied airspace, over densely populated areas typical of urban scenarios.

The foreseen large-scale sUAV operations are not currently possible without drastically reducing safety levels of low-altitude airspace and a global need for new concepts and enabling technologies is clearly identified in the aeronautical community. These needs find influential formulations in NASA’s UAS traffic management [2] (UTM) and the European Commission’s U-space visions [3]. All the paradigms proposed so far assume a range of capabilities at an increasing level of autonomy, including Beyond Visual Line of Sight (BVLOS) operations, interactive planning, de-conflict operations with geo-fencing, collision and obstacle avoidance. A common element to these capabilities is the necessity to estimate the position and velocity of each vehicle.

sUAV navigation is typically based on the integration of low-cost Global Navigation Satellite System (GNSS) receivers and commercial-grade Micro-Electro-Mechanical Systems (MEMS)-based inertial and magnetic sensors [4,5]. In nominal conditions, these navigation systems can provide an accuracy of approximately 5–10 m [6], good enough to implement, many autonomous functionalities. In urban environments, this accuracy can be hindered by the presence of obstacles, or greater accuracy may be required in order to perform special operations.

This article deals with sUAV position and velocity estimation within the UTM context. If, on the one hand, the peculiar characteristics of a UTM scenario can be seen as a source of open issues to be faced with, if appropriately interpreted, on the other hand, it is possible to derive benefits compared to traditional estimation methods in terms of accuracy, availability and continuity. In fact, the high density of traffic, the presence of numerous and heterogeneous on-board sensors (in addition to GNSS and inertial and magnetic sensors, low-cost vision-based systems [7,8] and micro radars [9] are becoming increasingly widespread), the presence of vehicle-to-vehicle communication channels are all opportunities to be exploited.

The basic idea is simple and intuitive in nature. The presence of numerous on-board sensors provides a great deal of information on the state of each vehicle. The measurements of certain sensors, such as vision-based systems or radars, contain information not only regarding the vehicle hosting the sensors, but also related to other vehicles (e.g., relative distance). The exchange of information between neighboring vehicles can allow them to improve the estimates of their position and velocity. A typical condition that can benefit from this situation is the navigation of vehicles in an GNSS-denied zone, where position and velocity can be estimated thanks to relative position sensors with other vehicles flying nearby.

The idea is currently being widely investigated. In [10,11,12], a multiple vehicle configuration is proposed to improve navigation (and attitude estimation) performance of a chief vehicle exploiting differential GPS using information deriving from a formation of flying deputy vehicles. In [13,14,15], a GPS-denied condition is specifically addressed in similar multi-vehicle configurations. These works consider a fixed number of vehicles flying in formation. A peculiar feature of the UTM scenario is the non-preordained motion of the vehicles, free to fly in the airspace and the absence of hierarchical relationships between the vehicles, which must all have access to the same minimum navigation performance.

The objective of the article is to describe a novel methodology applicable to the navigation of a sUAS fleet, which exploits the typical features of the UTM system in order to improve performance with respect to traditional methods. Therefore, the fundamental assumptions characterizing the considered scenario include the absence of a Central Processing Unit (CPU) for information elaboration or distribution (decentralization), or of a vehicle that is hierarchically distinct from the others. Moreover, only locally relevant computation is required to take place in each local processing unit, allowing the number of nodes to grow arbitrarily, without exceeding local computational resources (scalability). Thus, the starting point is to translate the basic idea of multi-sensor multi-vehicle exploitation into formal terms, with an algorithm preserving optimality characteristic typical of many common State Estimators.

The Kalman Filter (KF) represents the cornerstone for optimal state estimation. In its classical implementation [16], it has an intrinsically-centralized structure, in which the CPU samples the measurements and performs the estimation process. Although possibly optimal, when applied to Large-Scale Multi-Sensor Systems, it does not provide a solution compliant with the previously discussed assumptions. The main Centralized Kalman Filter limitations are the high computational load overcharging the CPU when the size of the system increases and the high communication complexity when the spatial distribution of the system expands.

In order to guarantee the Kalman Filter adequate scalability and decentralization characteristics, many decentralized algorithms [17,18,19,20,21,22,23,24] have been proposed, based on multiple local Kalman Filters, one in each local processing unit. In order to take processing and communication limits into account, the local Kalman Filters must involve the computation and communication only of local quantities of dimension $n_{l}\ll n,\;$ where *n* is the dimension of the global system.

Much of the existing studies [25,26,27,28] focus on large sensor network monitoring low-dimensional systems, and they mainly address the problem of how efficiently the available information is distributed. These solutions address scalability mainly looking to the dimension of the measurement signals, and not of the state of the system itself.

In other works [29,30], a reduced order Kalman Filters models have been proposed to specifically address the computation burden that arises from increasing the order of the global system. This research and other similar works address the issue of scalability in particular for fully connected or almost fully connected topologies. The algorithms based on this type of topology require long distance communication that reduces some of the benefits of decentralized architectures.

In order to address the problem over arbitrary communication networks, data fusion algorithms based on the Consensus Theory [31] are widely discussed in the literature. At the basis of consensus-based methodologies, there is the concept of covariance intersection [32], which represents a preliminary solution to the problem of merging the local estimates, so as to obtain a more accurate global estimate. All consensus-based methodologies can be interpreted as a generalization of the covariance intersection fusion rule. Consensus-based methodologies are iterative in nature. At the first iteration step, they conceptually correspond to the covariance intersection rule. When the number of iteration steps go to infinity, under certain conditions, they tend towards the centralized solution. This is a highly desirable characteristic.

A first form of Consensus-based algorithm for linear systems is the Consensus on Information (CI), discussed in [33]. The methodology derives from a decentralized estimation algorithm with stability properties guaranteed under collective observability and network-strong connectivity (thus ensuring a relaxation of the condition of full connectivity). Although these stability characteristics are guaranteed, even for only one single consensus step (in this case, the rule corresponds to the covariance intersection), the results obtained applying algorithms of this family do not tend towards a centralized solution for a number of consensus steps that tend towards infinity.

A different approach of Consensus-based estimation for linear systems is named Consensus on Measurements (CM). This method is discussed, among others, in [34,35], and differs from the CI for the quantities on which the consensus procedure is carried out. Unlike CI, Consensus on Measurements tends towards the equivalent centralized solution as the number of consensus steps goes to infinity, but does not guarantee stability unless the number of consensus step is sufficiently high.

In [36], a hybrid consensus approach is described, defined by the author as the Hybrid Consensus on Measurement and on Information (HCMCI), based both on CM and CI. The scope of the proposed approach is to combine their complementary benefits avoiding their main drawbacks. The HCMCI algorithm, which, among other things, extends the consensus-based solution to the non-linear case using an Extended Kalman Filter approach, appears to be a promising methodology for dealing with the problem of the distribution of information in systems of a more general topology.

Consensus-based methods address the issue of decentralization of the estimate by reducing the complexity of the communication system even in the case of systems that are not fully connected. However, these algorithms do not address the problem of scalability: in all the aforementioned methods, each local model has a cardinality equal to that of the global system. These solutions, therefore, do not address the problem of scalability as the size of the system grows.

Furthermore, a common element of consensus-based approaches is the stability criterion of the solution related to the system topology. Strong connectivity is a required condition. Without going into formalisms, this translates into the assumption that each vehicle is connected to the others at least through an indirect route, passing through the other vehicles. If the stability conditions, specifically relating to the system topology, continue to be verified in subsets, the proposed algorithms can be used locally. The topology of a system such as a fleet of sUAVs free to fly in space is highly variable over time. Therefore, it would be advantageous to have an appropriate clustering mechanism that consists of locally stable estimation systems formed of only a properly selected global state subset.

The methodology proposed in this article combines the results achieved for scalable decentralized systems obtained by the Internodal Transformation Theory methodology [29], with the advantages guaranteed by the use of consensus-based techniques [31]. The goal is to inherit the scalability properties from the former, the ability to distribute information within the strongly connected sub graphs from the latter, and to obtain, from the combined use of the two approaches, a self-clustering property, through which the local elements that form the global system self-aggregate, in order to form sub-systems in which the solution is stable.

## 2. Materials and Methods

This section introduces the fundamental notions for the definition of a decentralized estimation algorithm. Subsequently, it introduces the basic concepts of both the Internodal Transformation Theory and the Consensus Theory necessary for the proposed algorithm formulation. Finally, it defines the algorithm in its generic formulation identifying a possible application to the problem of estimating position and velocity for a sUAV fleet.

### 2.1. Problem Formulation

This article addresses Scalable Distributed State Estimation (SDSE) over a network consisting of nodes representing free-flying vehicles. Each vehicle has its own on-board sensors that can locally process sensor data and exchange data with other vehicles. The problem can be expressed as follows.

Let us consider a set of *N* flying vehicles. Communication topology between vehicles at any time *k* can be defined in terms of a directed graph G={V,A}, where $V=\left\{V_{1},\ldots V_{N}\right\}\;$is the set of vehicles (nodes of the graph) and A is the set of pair describing a communication link from $V_{i}$ to$\;V_{j}$ (arcs): vehicle *i* can receive data from vehicle *j* if $\left(V_{i},V_{j}\right)\in A$. For each vehicle *i*, let be $A^{i}=\left\{V_{j}\in V:\left(V_{i},V_{j}\right)\in A\right\}$ the set of its neighbors.

Let x∈ℝn be the global system state. State x is global in that it includes the information to describe the behavior of each vehicle.

Let us consider a non-linear dynamic model of the system in the state space form and discrete time domain:(1)xk=f(xk−1)+ωk−1
where f is the non-linear state transition function and ωk∈ℝn is the process noise. For each vehicle *i,* let us consider a set of non-linear measurements zki∈ℝm given by:(2)zki=hi(xk)+νki    Vi∈V 
where hi is the local observation model and νi∈ℝm is the measurement noise. Let us assume that the process and measurement noise ω and νi are mutually uncorrelated zero-mean noise with covariance Ωk−1=E[ωk−1ωk−1T]>0 and Rki=E[νkiνkiT]>0.

The objective of a state estimation problem is to have, at each time *k* an estimation of the global system state$\;{\hat{x}}_{k}\;$based on measurement$\;z_{k}={\left[z_{k}^{1T},\ldots ,z_{k}^{NT}\right]}^{T}$.

An SDSE problem introduces the concepts of Distribution and Scalability:In a Distributed State Estimation, each node has a locally-processed estimation of the global system state ${\hat{x}}_{k}^{i}$ based only on the local measurements $z_{k}^{i}\;$and date received form the adjacent vehicles j, such that $V_{j}\in A^{i}$
In a Scalable State Estimation, a model distribution logic reduces the size of local estimation problems allowing each node to estimate only a subset of the global state.


### 2.2. The Basis of Decentralization

To address the problem, a fundamental result relating to the centralized problem can be briefly recalled. First, let us consider the overall system given the state dynamics (1) and the overall measurement model:(3)$$z_{k}=h\left(x_{k}\right)+\nu_{k}$$
where $\nu_{k}={\left[\nu_{k}^{1T},\ldots ,\nu_{k}^{NT}\right]}^{T}$.

The estimation of the state x at time k, given information up to and including time (k−m) and the corresponding variance ***P*** are given by:(4)$${\hat{x}}_{k|k-m}=E[\;x_{k}\;\;\;|z_{1},\ldots z_{k-m}\;]$$
(5)$$P_{k|k-m}=E\left[{\left(x_{k}-{\hat{x}}_{k|k-m}\right)}^{T}\;\left(x_{k}-{\hat{x}}_{k|k-m}\right)\;|z_{1},\ldots z_{k-m}\right]$$
being, in a recursive formulation, the most relevant cases those for *m* = *k*−1 and *m = k*.

The information filter, equivalent to the traditional Covariance Kalman Filter, provides a recursive estimation ${\hat{x}}_{k|k}$ for the state x at time *k* given the information $z_{1},\ldots z_{k}$ up to time *k*. The information matrix Q and information state vector q can be defined [37] as the inverse of covariance matrix and as the product of the inverse of the covariance matrix and the state estimate, respectively:(6)$$Q_{k|k-m}=P_{k|k-m}^{-1}$$
(7)$$q_{k|k-m}=Q_{k|k-m}{\hat{x}}_{k|k-m}$$

In terms of information space variables, the Extended Information Kalman Filter can be written in the following form [29], given without derivation:prediction
(8)$${\hat{x}}_{k|k-1}=f\left({\hat{x}}_{k-1|k-1}\right)$$
(9)$$Q_{k|k-1}={\left[\frac{\partial f\left({\hat{x}}_{k-1|k-1}\right)}{\partial x_{k}}\;\;\;Q_{k-1|k-1}^{-1}\;\;\;{\frac{\partial f\left({\hat{x}}_{k-1|k-1}\right)}{\partial x_{k}}}^{T}+\Omega_{k-1}\right]}^{-1}$$
(10)$$q_{k|k-1}=Q_{k|k-1}\;{\hat{x}}_{k|k-1}$$
correction
(11)$$q_{k|k}=q_{k|k-1}+i_{k}$$
(12)$$Q_{k|k}=Q_{k|k-1}+I_{k}$$
where
(13)$$I_{k}={\frac{\partial h\left({\hat{x}}_{k|k-1}\right)}{\partial x_{k}}}^{T}{\left(R_{k}\right)}^{-1}\frac{\partial h\left({\hat{x}}_{k|k-1}\right)}{\partial x_{k}}$$
(14)$$i_{k}={\frac{\partial h\left({\hat{x}}_{k|k-1}\right)}{\partial x_{k}}}^{T}{\left(R_{k}\right)}^{-1}\;\left[c_{k}+\frac{\partial h\left({\hat{x}}_{k|k-1}\right)\;}{\partial x_{k}}{\hat{x}}_{k|k-1}\right]$$
(15)$$c_{k}=z_{k}-h\left({\hat{x}}_{k|k-1}\right)$$

The Global Information Filter applies to a fully centralized fusion architecture composed of a central processing unit directly connected to all sensing devices. One of the major advantages of the information filter formulation is its capability to be easily decentralized into a network of communicating nodes. In a decentralized fusion architecture, the system consists of different processing nodes able to perform the estimation of the global state on the basis of local observation and possible shared observations coming from other nodes. Let us first consider the case of a network of fully connected nodes *N*, which supposes that each vehicle is connected to all the other vehicles. Let us assume that each local node has a state model identical to the centralized model (1), i.e., each node performs a global state estimation x^i∈ℝn. Local information matrix and information state vectors can be defined in each node *i:*(16)Qk|k−mi=(Pk|k−mi)−1 
(17)qk|k−mi=Qk|k−mix^k|k−mi

The estimation algorithm for node *i* with information being communicated to it from other *(N-1)* nodes can be expressed in the following step analogous to the centralized case:prediction
(18)$${\hat{x}}_{k|k-1}^{i}=f\left({\hat{x}}_{k-1|k-1}^{i}\right)$$
(19)$$Q_{k|k-1}^{i}={\left[\frac{\partial f\left({\hat{x}}_{k-1|k-1}^{i}\right)}{\partial x_{k}}\;\;\;\left(Q_{k-1|k-1}^{i}\right){\;}^{-1}\;\;{\frac{\partial f\left({\hat{x}}_{k-1|k-1}^{i}\right)}{\partial x_{k}}}^{T}+\Omega_{k-1}^{i}\right]}^{-1}$$
(20)$$q_{k|k-1}^{i}=Q_{k|k-1}^{i}\;{\hat{x}}_{k|k-1}^{i}$$
correction
(21)$$q_{k|k}^{i}=q_{k|k-1}^{i}+\overset{N}{\underset{j=1}{\sum }}i_{k}^{j}$$
(22)$$Q_{k|k}^{i}=Q_{k|k-1}^{i}+\overset{N}{\underset{j=1}{\sum }}I_{k}^{j}$$
where
(23)$$I_{k}^{j}={\frac{\partial h^{j}\left({\hat{x}}_{k|k-1}^{j}\right)}{\partial x_{k}}}^{T}{\left(R_{k}^{j}\right)}^{-1}\frac{\partial h^{j}\left({\hat{x}}_{k|k-1}^{j}\right)}{\partial x_{k}}\;$$
(24)$$i_{k}^{j}={\frac{\partial h^{j}\left({\hat{x}}_{k|k-1}^{j}\right)}{\partial x_{k}}}^{T}{\left(R_{k}^{j}\right)}^{-1}\;\left[c_{k}^{j}+\frac{\partial h^{j}\left({\hat{x}}_{k|k-1}^{j}\right)\;}{\partial x_{k}}{\hat{x}}_{k|k-1}^{j}\right]$$
(25)$$c_{k}^{j}=z_{k}^{j}-h^{j}\left({\hat{x}}_{k|k-1}^{j}\right)$$

In the correction step expressed it can be assumed that each node begins with a common initial information state (e.g., $q_{0}^{i}=0\;Q_{0}^{i}=0\;\forall \;i$). The summations in Equations (21) and (22) are feasible because of the full connectivity of the system. Under these conditions, each local estimate is identical to the centralized system defined by the Equations (8)–(15).

The case of a fully connected system does not guarantee any real advantages with respect to the centralized case, both in terms of computational burden and of communication requirements. Nevertheless, it represents a starting point for the definition of any decentralized estimation algorithm based on a Kalman Filter.

### 2.3. Scalability

Let us first deal with the problem of scalability. In order to obtain the desired scalable solution, let us introduce the model distribution concepts as defined in [29]. Let us consider a local state for the node *i* related to the global state at time instant *k* by:(26)$$x_{k}^{i}=T_{k}^{i}x_{k}$$
where $T_{k}^{i}\;$is a linear nodal transformation matrix that select states or linear combinations of states from the global state vector. Using the Internodal Transformation Theory, it is possible to obtain a formulation in which each node performs the same estimation as the centralized formulation for a subset of the global state, while minimizing communication between nodes.

In order to derive a scalable solution, the inverse operation to model reduction is required. Generally speaking, $T_{k}^{i}$ is rectangular and its ordinary inverse is not defined. Hence, the use of the generalized pseudo-inverse ${\left(T_{k}^{i}\right)}^{+}$ is required. The pseudo-inverse provides the solution to the problem of reconstructing the global state $x_{k}$ starting from the local state $x_{k}^{i}\;$in node *i*, minimizing $∥x_{k}^{i}-T_{k}^{i}x_{k}∥$:(27)$$x_{k}={\left(T_{k}^{i}\right)}^{+}x_{k}^{i}$$

The geometrical interpretation of the reconstructed global state is a vector containing an unscaled relevant state and a zero in place of any irrelevant state to node *i*.

Let us introduce the concepts of information contribution at node *i* due to current observation from node *j,* defined as $i_{k}^{i|j}$, and the associated local information matrix $I_{k}^{i|j}$.

Error covariance at node *i* based on local observation from node *j* can be defined as:(28)$$P_{k|k}^{i|j}={\left(I_{k}^{i|j}\right)}^{+}\;$$
and the corresponding local estimate at node *i* based only on local observation from node *j* can be obtained from:(29)$${\hat{x}}_{k|k}^{i|j}=P_{k|k}^{i|j}i_{k}^{i|j}\;$$

It is possible to rewrite [29] the distributed formulation of Equations (18)–(25) in an equivalent scalable form in which each node propagates only locally relevant states and exchanges only relevant information with any other node:Prediction
(30)$${\hat{x}}_{k|k-1}^{i}=T_{k}^{i}f\left({\left(T_{k-1}^{i}\right)}^{+}{\hat{x}}_{k-1|k-1}^{i}\right)$$
(31)$${\tilde{Q}}_{k-1|k-1}^{i}=T_{k}^{i}\left\{{\left(T_{k-1}^{i}\right)}^{+}Q_{k-1|k-1}^{i}\left(T_{k-1}^{i}\right)\right\}{\left(T_{k}^{i}\right)}^{+}$$
(32)$$Q_{k|k-1}^{i}={\left[\frac{\partial f\left({\hat{x}}_{k-1|k-1}^{i}\right)}{\partial x_{k}}\;\;\;\left({\tilde{Q}}_{k-1|k-1}^{i}\right){\;}^{-1}\;\;{\frac{\partial f\left({\hat{x}}_{k-1|k-1}^{i}\right)}{\partial x_{k}}}^{T}+\Omega_{k-1}^{i}\right]}^{-1}$$
(33)$$q_{k|k-1}^{i}=Q_{k|k-1}^{i}{\hat{x}}_{k|k-1}^{i}$$
Correction
(34)$$q_{k|k}^{i}=q_{k|k-1}^{i}+\overset{N}{\underset{j=1}{\sum }}i_{k}^{i|j}$$
(35)$$Q_{k|k}^{i}=Q_{k|k-1}^{i}+\overset{N}{\underset{j=1}{\sum }}I_{k}^{i|j}$$

In the correction phase (34), (35), it is assumed that each node *i* receives from the other nodes the local information $\left(i_{k}^{i|j},\;I_{k}^{i|j}\right)$. Each node is able to calculate the information contributions locally starting from local measures without relying on information communicated by the other nodes in a completely analogous way to what was done in Equations (23)–(25):(36)$$i_{k}^{i|i}={\frac{\partial h^{i}\left({\hat{x}}_{k|k-1}^{i}\right)}{\partial x_{k}}}^{T}{\left(R_{k}^{i}\right)}^{-1}\;\left[c_{k}^{i}+\frac{\partial h^{i}\left({\hat{x}}_{k|k-1}^{i}\right)\;}{\partial x_{k}}{\hat{x}}_{k|k-1}^{i}\right]$$
(37)$$I_{k}^{i|i}={\frac{\partial h\left({\hat{x}}_{k|k-1}^{i}\right)}{\partial x_{k}}}^{T}{\left(R_{k}^{i}\right)}^{-1}\frac{\partial h\left({\hat{x}}_{k|k-1}^{i}\right)}{\partial x_{k}}$$

It is therefore necessary to look for transformations that locally carry out the following transformations in each node.
(38)$$I_{k}^{j|j}\to I_{k}^{i|j}\;\;\;\;\;\;\;\forall j\neq i$$
(39)$$i_{k}^{j|j}\to i_{k}^{i|j}\;\;\;\;\;\;\;\;\forall j\neq i$$

It is possible to show [29] that the Information Space Intermodal Transformation map can be schematized as in Figure 1, where:(40)$$V_{k}^{ji}=T_{k}^{i}\;\;{\left(T_{k}^{j}\right)}^{+}$$
(41)$$T_{k}^{ji}=I_{k}^{i|j}V_{k}^{ji}{\left(I_{k}^{j|i}\right)}^{+}$$

The information parameter in node $V_{i}$ given only node $V_{j}$ observation $z_{k}^{j}$ can thus be been derived in each node starting from quantities, calculated locally as follows
(42)$$I_{k}^{i|j}={\left[T_{k}^{i}\;\;{\left[T_{k}^{j}{}^{T}\left(\;I_{k}^{j|j}\;\right)\;T_{k}^{j}\right]}^{+}T_{k}^{i}{}^{T}\right]}^{+}={ℱ}_{I}^{j\to i}\left(\;I_{k}^{j}\;\right)$$
(43)$$\;i_{k}^{i|j}=T_{k}^{ji}i_{k}^{j|j}={ℱ}_{i}^{j\to i}\left(\;i_{k}^{j|j}\;\right)$$

In order to carry out the transformations (42) and (43), each node *j* must therefore communicate the information to node *i*: $\left(T_{k}^{j},I_{k}^{j|j},i_{k}^{j|j}\right)$.

The solution identified, as highlighted by Equations (34) and (35), formally still applies to a fully connected system. The process of minimizing communication between nodes is highly dependent on the choice of the matrices $T_{k}^{i}$ of the model distributions. On the other hand, no hypothesis has been made so far about the criteria to be used to select these matrices. The effect of minimizing communications is evident by observing that $T_{k}^{ji}=T_{k}^{ij}=0$ if two nodes do not share any common state. Therefore, the exchange of any information is not necessary between two nodes not sharing any common state. It is possible to extend this consideration to two or more sub-graphs that are individually strongly connected yet which are not connected to each other. By choosing a local state for each node that includes only the states of nodes belonging to its strongly connected subgraph, the need for communication between unconnected sub graphs is avoided. The selected algorithm then performs a sort of clustering of the estimation process selecting groups of nodes that require exchanging data (see Figure 2).

### 2.4. Consensus Based Information Distribution

The algorithm discussed in the previous paragraph does not completely solve the problem of the information distribution: two nodes that share part of the local state must exchange information directly. In this way, the algorithm autonomously manages the formation and disintegration of connected sub-graphs, but a fully connectivity is required in each sub-graph. To overcome this problem, it is possible to use the Consensus Theory.

Let us consider a strongly connected subgraph $\tilde{G}=\left\{\tilde{V},\tilde{A}\right\}\subseteq G$ included in the global graph. The summation of generic terms $\chi_{i}$ distributed between its nodes:(44)$$X=\underset{i\in \tilde{A}}{\sum }\chi_{i}$$
can be obtained with a consensus averaging iterative process [38]:$$\forall \;V_{i}\in \tilde{A}\;\;\;\;\;\chi_{i}^{\left(0\right)}=\chi_{i}$$
(45)$$for\;\ell =1,\ldots ,L\;\;\;\;\;\;\;\chi_{i}^{\left(\ell +1\right)}=\underset{j\in \tilde{A}}{\sum }\mu_{ij}\chi_{j}^{\left(\ell \right)}$$
(46)$$X=\tilde{N}\chi_{i}^{\left(L\right)}$$
where $\tilde{N}$ is the number of nodes of the strongly connected sub-graph.

With a proper choice of the $\mu_{ij}$, in fact the solution converges to the average vector
(47)$$\underset{\ell \to \infty }{l\mathrm{im}}\chi_{i}^{\left(\ell \right)}=\frac{1}{\tilde{N}}\underset{j\in A_{i}}{\sum }\chi_{j}\;\;\;\;\;\;\forall V_{i}\in \tilde{A}$$

A possible choice of terms $\mu_{ij}$ is to select them as local-degree weights:(48)$$\mu_{ij}=\frac{1}{max\left\{d\left(i\right),d\left(j\right)\right\}}\;,\;\;\;\left(V_{i},V_{j}\right)\in A$$
(49)$$\mu_{ij}=0,\;\;\;\left(V_{i},V_{j}\right)\notin A$$
(50)$$\mu_{ii}=1-\underset{j\in A_{i}}{\sum }\mu_{ij}$$
where *d(i)* is the degree of the node $V_{i}$.

### 2.5. Algorithm Description

By applying to the Scalable Distributed algorithm defined by Equations (30)–(35), a consensus procedure to asymptotically obtain the summations in Equations (34) and (35), and adding a consensus procedure on a priori information pair to reproduce a Hybrid Consensus on Measurement and on Information (HCMCI) formulation [36], the following algorithm can be obtained:
$$\mathrm{Update}T_{k}^{i}$$
Local prediction
(51)$${\hat{x}}_{k|k-1}^{i}=T_{k}^{i}f\left({\left(T_{k-1}^{i}\right)}^{+}{\hat{x}}_{k-1|k-1}^{i}\right)$$
(52)$${\tilde{Q}}_{k-1|k-1}^{i}=T_{k}^{i}\left\{{\left(T_{k-1}^{i}\right)}^{+}Q_{k-1|k-1}^{i}\left(T_{k-1}^{i}\right)\right\}{\left(T_{k}^{i}\right)}^{+}$$
(53)$$Q_{k|k-1}^{i}={\left[\frac{\partial f\left({\hat{x}}_{k-1|k-1}^{i}\right)}{\partial x_{k}}\;\;\;\left({\tilde{Q}}_{k-1|k-1}^{i}\right){\;}^{-1}\;\;{\frac{\partial f\left({\hat{x}}_{k-1|k-1}^{i}\right)}{\partial x_{k}}}^{T}+\Omega_{k-1}^{i}\right]}^{-1}$$
(54)$$q_{k|k-1}^{i}=Q_{k|k-1}^{i}\;{\hat{x}}_{k|k-1}^{i}$$
Consensus (oninformation) ∀ i
Initialization
(55)$$Q_{k|k-1}^{i}{}^{\left(0\right)}=Q_{k|k-1}^{i}\;q_{k|k-1}^{i}{}^{\left(0\right)}=q_{k|k-1}^{i}$$
for  ℓ=0,1,….L
$$\forall \;\;V_{j}\in A^{i}\;\mathrm{receive}\;Q_{k|k-1}^{j}{\;}^{\left(\ell \right)},\;q_{k|k-1}^{j}{}^{\left(\ell \right)}\;$$
(56)$$Q_{k|k-1}^{i}{\;}^{\left(\ell +1\right)}=\underset{\;j\in A_{i}}{\sum }\mu_{ij\;}\;\;\;T_{k}^{i}\left\{{\left(T_{k}^{j}\right)}^{+}Q_{k|k-1}^{j}{\;}^{\left(\ell \right)}\left(T_{k}^{j}\right)\right\}{\left(T_{k}^{i}\right)}^{+}$$
(57)$$q_{k|k-1}^{i}{\;}^{\left(\ell +1\right)}=\underset{\;j\in A_{i}}{\sum }\mu_{ij\;}\;\;T_{k}^{i}\;{\left(T_{k}^{j}\right)}^{+}q_{k|k-1}^{j}{\;}^{\left(\ell \right)}$$
Localmeasure ∀ i
$$\mathrm{Sample}z_{k}^{i}$$
(58)$$c_{k}^{i}=z_{k}^{i}-h^{i}\left({\hat{x}}_{k|k-1}^{i}\right)$$
(59)$$I_{k}^{i|i}={\frac{\partial h^{i}\left({\hat{x}}_{k|k-1}^{i}\right)}{\partial x_{k}}}^{T}{\left(R_{k}^{i}\right)}^{-1}\frac{\partial h^{i}\left({\hat{x}}_{k|k-1}^{i}\right)}{\partial x_{k}}$$
(60)$$i_{k}^{i|i}={\frac{\partial h^{i}\left({\hat{x}}_{k|k-1}^{i}\right)}{\partial x_{k}}}^{T}{\left(R_{k}^{i}\right)}^{-1}\;\left[c_{k}^{i}+\frac{\partial h^{i}\left({\hat{x}}_{k|k-1}^{i}\right)\;}{\partial x_{k}}{\hat{x}}_{k|k-1}^{i}\right]$$
Consensus (onmeasurement) ∀ i
Initialization
(61)$$I_{k}^{i}{}^{\left(0\right)}=I_{k}^{i}\;\;i_{k}^{j}{}^{\left(0\right)}=i_{k}^{j}$$
for  ℓ=0,1,….L
$$\forall \;\;V_{j}\in A^{i}\;receive\;I_{k}^{j|j}{\;}^{\left(\ell \right)},i_{k}^{j|j}{\;}^{\left(\ell \right)}\;$$
(62)$$I_{k}^{i|i}{\;}^{\left(\ell +1\right)}=\underset{\;j\in A_{i}}{\sum }\mu_{ij\;}\;\;\;{ℱ}_{I}^{j\to i}\left(\;I_{k}^{j|j}{\;}^{\left(\ell \right)}\right)$$
(63)$$i_{k}^{i|i}{\;}^{\left(\ell +1\right)}=\underset{\;j\in A_{i}}{\sum }\mu_{ij\;}\;\;\;{ℱ}_{i}^{j\to i}\left(\;i_{k}^{j|j}{\;}^{\left(\ell \right)}\right)$$
Correction ∀ i
(64)$$Q_{k|k}^{i}=Q_{k|k-1}^{i}+N^{i}I_{k}^{i|i}{\;}^{\left(L\right)}$$
(65)$$q_{k|k}^{i}=q_{k|k-1}^{i}+N^{i}i_{k}^{i|i}{\;}^{\left(L\right)}$$
(66)$${\hat{x}}_{k|k}^{i}={\left(Q_{k|k}^{i}\right)}^{-1}\;q_{k|k}^{i}$$


The logical building process of the Scalable DSE Algorithm is shown in Figure 3, together with the relationships with the algorithms from which it is derived.

The algorithm applies to a set of cooperating nodes (vehicle), in the sense that each node actively communicates information to neighboring nodes. The clustering property of the algorithm causes each node to estimate an automatically selected subset of the global state, based on the communication topology.

The mode selection update step can be approached in various ways. A simple way is based on a consensus procedure carried out on a local defined adjacency matrix initialized, in each node, as if the node were an isolated node.
(67)$$a_{hh}^{i}{}^{\left(0\right)}=1\;\;\;\;if\;h=i\;;\;otherwise\;\;a_{hk}^{i}{}^{\left(0\right)}=0$$
(68)$$for\;\ell =1,\ldots L\;\;\;\;\;\;A^{i}{}^{\left(\ell +1\right)}=\underset{j\in A_{i}}{\sum }\mu_{ij}A^{i}{}^{\left(\ell \right)}$$
where $A^{i}$ is a locally known adjacence matrix of elements$\;a_{hk}^{i}$. It is easy to verify that, for a proper number of consensus steps, the diagonal of each local adjacency matrix contains information on the nodes belonging to the same strongly connected sub graph (cluster).

The equations in which the nodal transformation matrices explicitly appear, Equations (51), (52), (56), (57), (62), (63), perform the mode distribution operations. From an implementation perspective these equations, apparently onerous from a computational point of view, in the case of our interest, in which the matrices $T_{k}^{j}$ make a simple selection of the states, are therefore reduced to rows and columns reordering and deletion operators. The topology of the system, and then the matrices $T_{k}^{j}\;$varies with time. When node is added to a cluster, Equations (51) and (52) include the initialization to null information pairs for the corresponding states. When a node is excluded from a cluster, Equations (51) and (52) correspond to the elimination of the corresponding rows and columns.

As far as the stability of the solution is concerned, the algorithm is equivalent to the algorithm described by Equations (30)–(35) for L→∞. On the other hand, the results achieved in the field of internodal transformations guarantee the equivalence of the local estimation and centralized state estimation limited to local retained states.

Stability can also be interpreted in a second way. During its evolution over time the system can be divided in clusters each strongly connected. Such clusters, as mentioned, varies over time through the disaggregation of a cluster or the fusion of two clusters. In each time interval in which a strongly connected cluster exists, the algorithm it is locally equivalent to an EFK algorithm, based on HCMCI consensus. Therefore, it is possible to consider locally the stability conditions applicable to that algorithm. These stability condition can be demonstrated in the linear time invariant form [36].

In the particular case that the following assumption are satisfied:The system is collectively observable;The network is strongly connected (i.e., any node is reachable from any other node through a directed path).
then, it is guaranteed that the estimation error is asymptotical bounded in mean square i.e., $\underset{k\to \infty }{\lim}\sup E\left\{∥{\hat{x}}_{k|k}^{i}-x_{k}^{i}∥{}^{2}\right\}<\infty$. The HCMCI formulation ensures convergence to the centralized EKF in each strongly connected cluster.

### 2.6. sUAV Positon Estimation

Let us consider a set of N co-operative flying vehicles$\;V=\left\{V_{1},\ldots V_{N}\right\}\;$free to move in a defined airspace of volume V. A vehicle $V_{I}$ can be located by means of its position $r^{i}=\left(x^{I},y^{I},z^{I}\right)$ in an inertial reference system *I*. Its attitude is defined in terms of Euler’s angles $\Theta^{i}\;$of a body reference frame B, with respect to inertial reference frame *I.*

In the proposed filter architecture, the chosen system dynamic model is a purely kinematic model not affected by any kind of model uncertainties. The state model equations for$\;V_{I}$ assumes the following form:(69)$${\dot{r}}^{i}=v^{i}$$
(70)$${\dot{v}}^{i}=M_{BI}^{i}\left(\Theta^{i}\right)f_{B}^{i}+G_{I}^{i}$$
where vector $v^{i}$ is the velocity vector in inertial reference system *I*; vector $f_{B}^{i\;}$ is the specific force (force per unit mass) vector expressed in body frame *B*; $G_{I}^{i}$ is the specific gravity force vector expressed in the inertial reference frame *I* and $M_{BI}^{i}$ represents the rotation matrix from *B* to *I*.

We assume each that vehicle is equipped with a sensor suite including three type of sensors:Inertial Measurement Unit sensors (3-axes gyros, accelerometers and magnetometers);Absolute position and velocity (e.g., GPS);Relative position (e.g., visual based sensors, radar, radio frequency of time of flight).

Using these three types of sensors, each vehicle can locally generate the set of measurement included in one or more of the following equations:(71)$$y_{acc}^{i}=f_{B}^{i}$$
(72)$$y_{ahrs}^{i}=\Theta^{i}$$
(73)$$y_{gnss}^{i}=\left[r^{i},v^{i}\right]$$
(74)$$y_{rel}^{i}=\left\{\left(R^{ij},\psi^{ij},\Phi^{ij}\right)\;,\;\;V_{j}\in {ℱ}^{i}\right\}$$

More specifically, Equation (71) represent the specific force vector measured by accelerometers. Furthermore, it can be assumed that each vehicle has a local AHRS (Attitude and Heading reference system) capable of estimating attitude angles, expressed in Equation (72), by properly filtering the measurements coming from an Inertial Measurement Unit (IMU) sensors.

Equation (73) represents GNSS measures of inertial position and velocity. Finally, Equation (74) is representative of the relative position measurement. It is constituted by a triple including relative distance R, elevation Φ and azimuth ψ of all vehicles included into to Field of View ${ℱ}^{i}\;$of the vehicle $V_{i}$. In this formulation, asynchronous measurements can occur both due to the different sampling times of the sensors, and for in and out Field of View transitions.

In order to complete the kinematic model, the following equations are added to the system: (69) and (70), to have an autonomous system capable to filter inertial sensors biases [39]:(75)$${\dot{f}}_{B}^{i}=0$$
(76)$${\dot{\Theta }}^{i}=0$$

Equations (75) and (76) are not restricting, due to the high update rates typical of the inertial measurements.

The state $\xi^{i}\;$of vehicle $V_{i}$ is described by:(77)$$\xi^{i}={\left[r^{i}{}^{T},v^{i}{}^{T},f_{B}^{i}{}^{T},\Theta^{i}{}^{T}\right]}^{T}$$

The overall state ξ of system is
(78)$$\xi ={\left[\xi^{1}{}^{T},\xi^{2}{}^{T},\ldots ,\xi^{N}{}^{T}\right]}^{T}$$

Finally, it can be assumed that each vehicle is equipped with a bi-directional communication device whose range defines the connection graph, according to the rule that a vehicle $V_{j}$ belong to neighbor $A^{i}$ of vehicle $V_{i}$ if its relative distance is less than the communication range.

## 3. Results

In order to highlight the most peculiar aspects of the proposed algorithm, which are those related to the decentralization, scalability and self-clustering of the local states in the nodes, the results are reported for a specific simplified, though sufficiently representative, scenario. The fundamental parameters, characteristic of the scenario, are summarized in Table 1. In order to facilitate the readability of the results, Equations (69), (70) are reduced to the two-dimensional case, i.e., two translational degrees of freedom and one angular degree of freedom. The measurements are asynchronous and obtained at different frequencies. The parameter Communication frequency in this context takes on the meaning of frequency of the consensus operation. The execution time of an entire consensus operation, consisting three Consensus steps (*L = 3*), is neglected.

Figure 4 shows the position of the fleet, consisting of five vehicles, in four different time instants. The blue circles around each vehicle indicate the range of the relative position sensors; the red lines connecting two vehicles highlight the presence of an active communication link. The points in red indicate the absence of the GNSS signal, the position covered by the signal is reported in green.

The coverage of the GNSS signal is distributed randomly with the only constraint of guaranteeing a ratio between GNSS denied areas and areas covered by the service equal to 50%. Even if it represents a simplified model, this distribution is representative of an urban canyon scenario.

The simulation shows a single cluster at time instant t = 5 s.; two clusters at time t = 10 s (Vehicle 4 remains separate from the others); three clusters at t = 25 s (Vehicles 4 and 2 separated from the others); once again, a unique strongly connected system at the time instant t = 30 s.

Table 2 summarizes, at the time instants shown in Figure 4, the states propagated from each vehicle. Within each cluster, collective observability is guaranteed by the presence of at least one vehicle with GNSS available [40].

Figure 5, Figure 6 and Figure 7 show the estimates made locally by Vehicle 1. The EKF reference (black line) is obviously reported only in the case of self-estimation. It corresponds to the estimate made by the vehicle using only its own on-board sensors. In cross-estimation, whenever a vehicle is not in the same cluster of the estimating vehicle, the curve representing the estimate is interrupted and then resumed as soon as that vehicle returns to the cluster.

Three fundamental behaviors can be observed:The self-clustering mechanism is evident. Vehicle 2 is disconnected from the estimate around t = 20 s (Figure 6). Vehicle 4 shows two estimation black out windows (Figure 7) around t = 6 sec and around t = 21 s. In both cases, the estimate is correctly recovered when the vehicle rejoins the cluster of the estimating vehicle. When this happens, the transient is adequately managed in the average process of Equations (56) and (57), in which the contribution of each vehicle is weighed with its own covariance matrix.The self-estimation of Vehicle 1 is more accurate than the traditional centralized EKF estimate. Vehicle 1 periodically loses the GNSS signal. This leads to a widespread degradation of the estimate. It is particularly evident around t = 20 s, in which the GNSS denied persists for a longer interval. In the decentralized solution, the observability of the cluster 1-3-5 is guaranteed by the GNSS signal received by Vehicle 3 (see Figure 4c).The estimation process is generally good, with the exception of the intervals of time when some vehicles do not share the same cluster. However, this situation represents the case in which these vehicles cannot communicate with the node making the estimate. In this case, the vehicles cannot share their data to improve their estimates This condition is obviously not a limitation of the algorithm.

Similar conclusions can be drawn for the estimates made by the other vehicle. For illustrative purposes only, estimates for Vehicle 4 are reported in Figure 8, Figure 9 and Figure 10.

## 4. Conclusions

The article discussed a solution to the problem of Distributed State Estimation (DSE) in the framework of Kalman Filter-based algorithms. The proposed solution draws from the theoretical results derived from two different methodologies both related to the Kalman Filter theory: Internodal Transformation Theory and Consensus Theory.

From the former, the algorithm inherits the scalability property, that is the ability to decompose the global problem into different reduced order problems on a local level. From the latter, it inherits the ability to efficiently distribute information among local sensing and computational nodes.

A novel property, deriving from the fusion of the two methodologies, is the self-clustering property of the nodes which aggregate themselves in local estimation sub problems in response to the variation of the communication topology. The aggregation process is performed by each node through the information dynamically transmitted by its neighboring nodes, and is achieved by reaching an agreement resulting from a Consensus-based process.

The proposed algorithm makes it possible to obtain more accurate estimates than those obtainable individually from each node that uses only local measurements. Furthermore, the scalability property reduces the computational burden in each node by means of reducing the size of the local problems, while decentralization improves the communication efficiency, allowing each vehicle to exchange information only to the nearest vehicle.

The algorithm has a general validity, but assumes a specific meaning if applied to set of co-operating sUAVs equipped with heterogeneous on-board sensors and limited range communication devices. In this perspective, the algorithm is proposed as a formalization of an intuitive concept in which vehicles flying nearby other co-operative vehicles share its own on-board measurement to enable a better estimate of each vehicle.

A particularly significant condition is, for example, that in which a vehicle flying in a GNSS-denied zone can use the measurements of its position transmitted from another vehicle (i.e., from a micro radar or from a vision-based system) to correct the estimate regarding its own position. This scenario represents, among others, typical operational conditions of a fleet of sUAVs in a UTM context, in which registered vehicles can perform free flight operations in a potentially highly density airspace in urban environment. The application of the algorithm to a problem that involves the presence of non-collaborating elements, such as intruders, may be subject to future works.

The numerical examples shown have no ambitions of a complete numerical validation, but want to represent a clear example of the benefits that the proposed algorithm can guarantee.

## Figures and Tables

**Figure 1 sensors-20-02682-f001:**
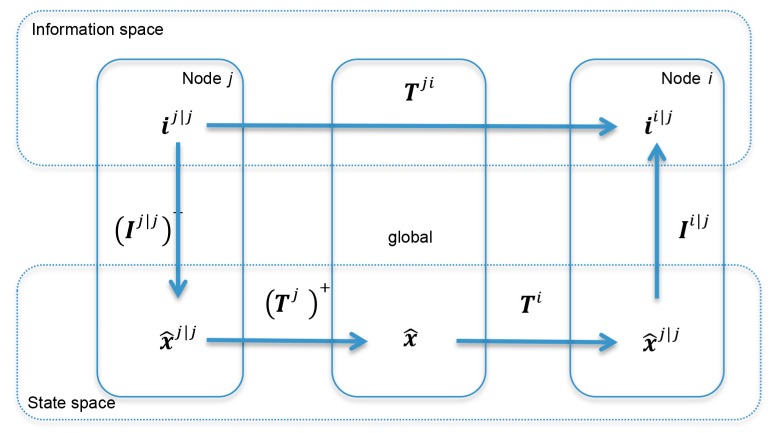
Information Space Intermodal Transformation map.

**Figure 2 sensors-20-02682-f002:**
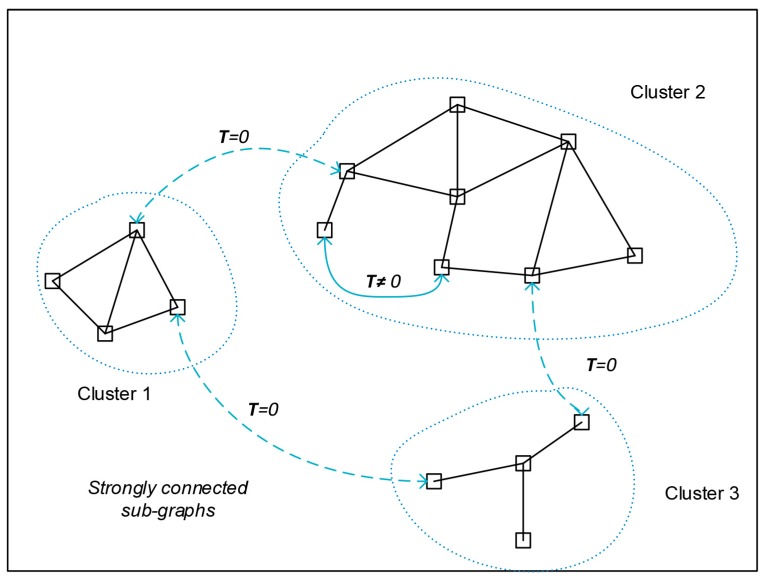
Internodal Transformation and graph connectivity.

**Figure 3 sensors-20-02682-f003:**
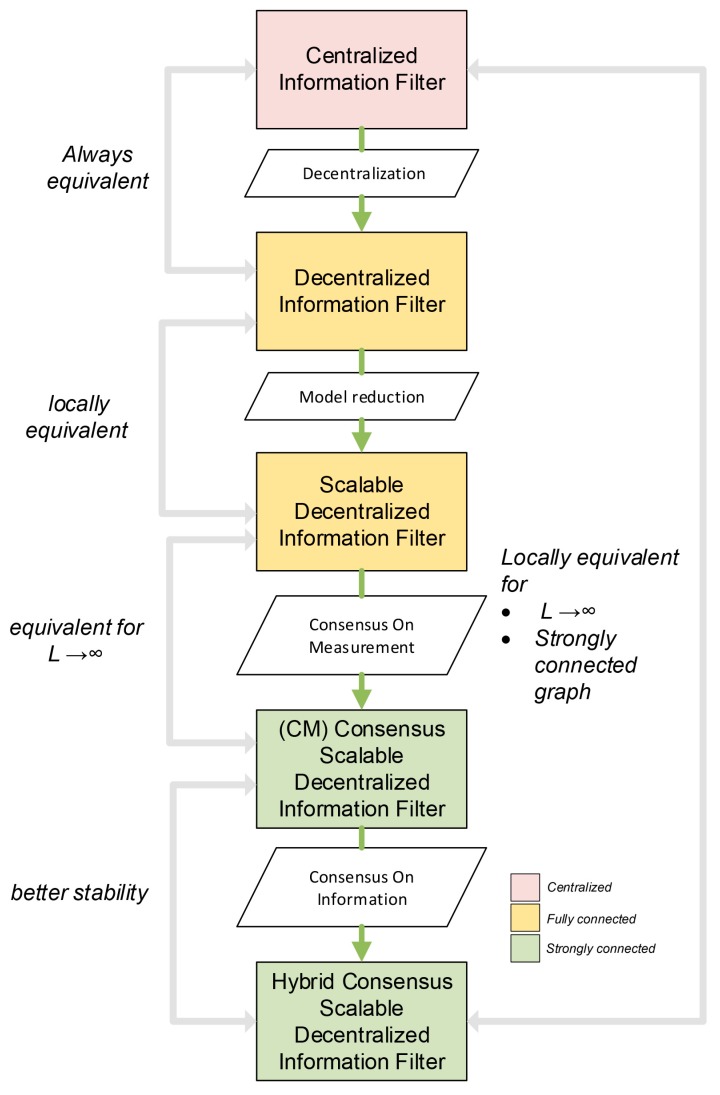
Scalable Distributed State Estimation (DSE) algorithm logical building process.

**Figure 4 sensors-20-02682-f004:**
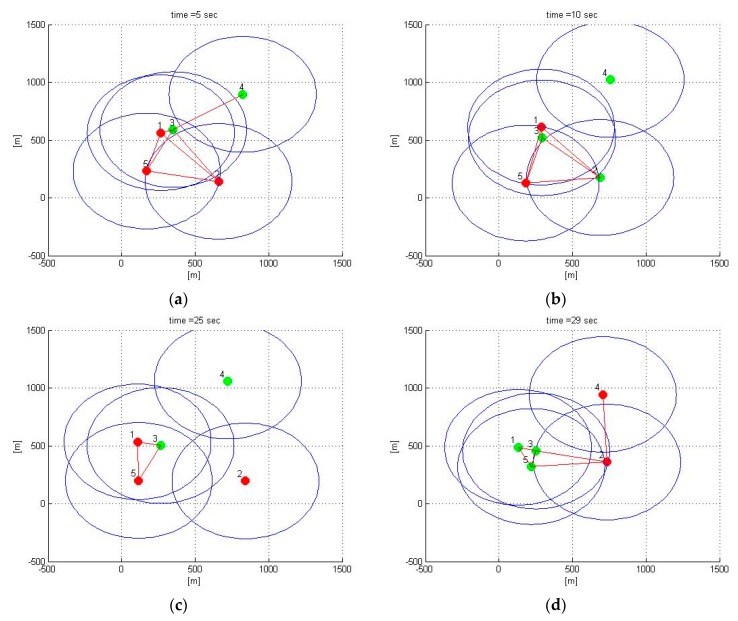
Fleet true position (**a**) t = 5 s (**b**) t = 10 s (**c**) t = 25 s (**d**) t = 30 s.

**Figure 5 sensors-20-02682-f005:**
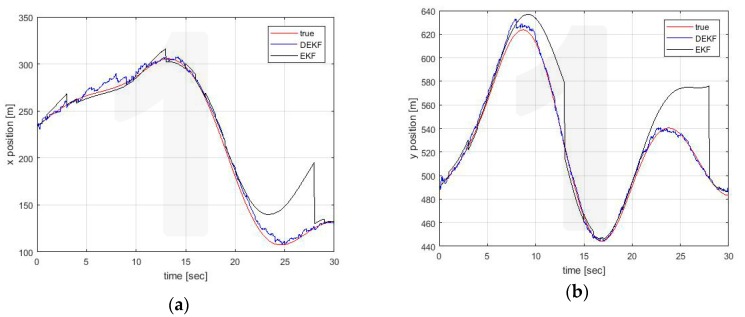
Position estimation performed by Vehicle 1 on its own position (**a**) x axis (**b**) y axis.

**Figure 6 sensors-20-02682-f006:**
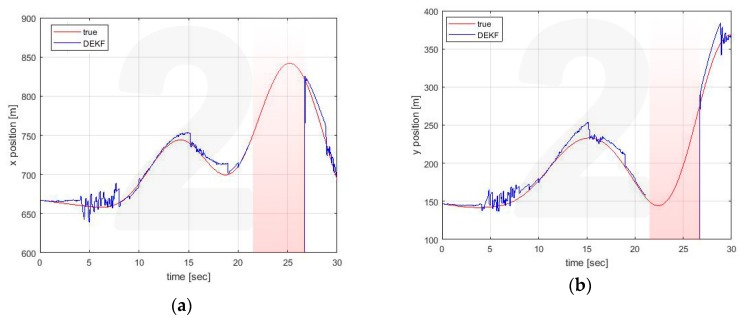
(**a**) Position estimation performed by Vehicle 1 on Vehicle 2 position (**a**) x axis (**b**) y axis.

**Figure 7 sensors-20-02682-f007:**
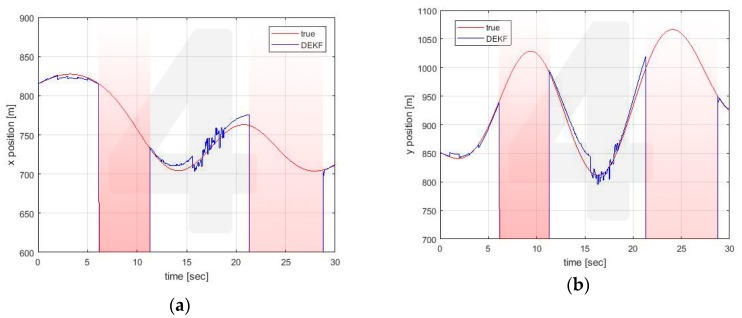
Position estimation performed by Vehicle 1 on Vehicle 4 position (**a**) x axis (**b**) y axis.

**Figure 8 sensors-20-02682-f008:**
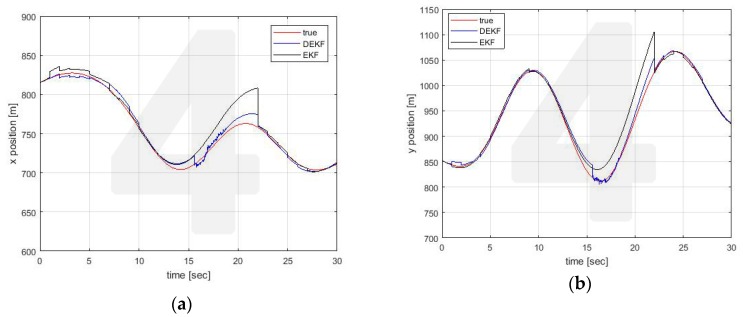
Position estimation performed by Vehicle 4 on its own position (**a**) x axis (**b**) y axis.

**Figure 9 sensors-20-02682-f009:**
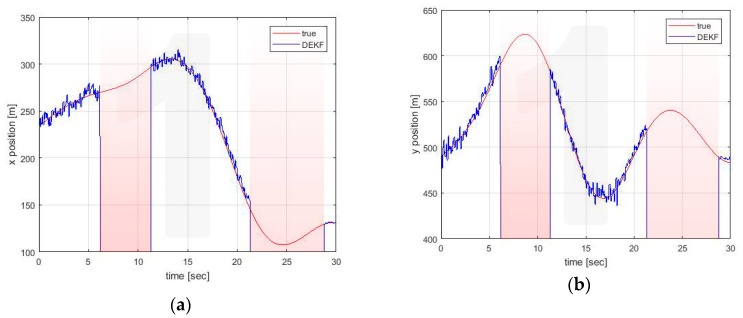
Position estimation performed by Vehicle 4 on Vehicle 1 position (**a**) x axis (**b**) y axis.

**Figure 10 sensors-20-02682-f010:**
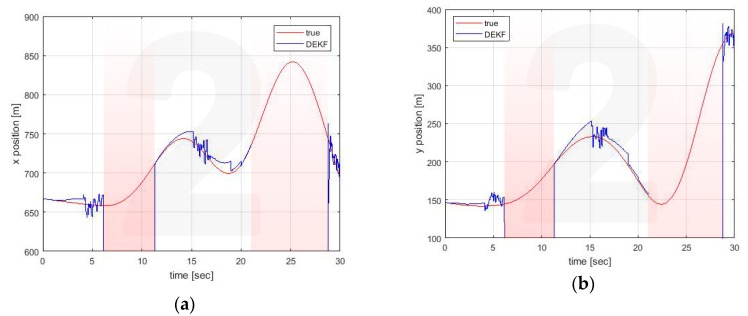
Position estimation performed by Vehicle 4 on Vehicle 2 position (**a**) x axis (**b**) y axis.

**Table 1 sensors-20-02682-t001:** Scenario description.

Simulation Parameter	Value
Number of Vehicles	5
GNSS coverage	50%
GNSS frequency	1 Hz
Radar Range	500 m
Radar frequency	1 Hz
IMU frequency	10 Hz
Communication Range	600 m
Communication frequency	1 Hz

**Table 2 sensors-20-02682-t002:** Model Distribution (estimated vehicle) for each vehicle.

	t = 5 s	t = 10 s	t = 25 s	t = 30 s
vehicle 1	1 2 **3 4**^1^ 5	1 **2 3** 5	1 **3** 5	**1** 2 **3** 4 **5**
vehicle 2	1 2 **3 4** 5	1 **2 3** 5	2	**1** 2 **3** 4 **5**
vehicle 3	1 2 **3 4** 5	1 **2 3** 5	1 **3** 5	**1** 2 **3** 4 **5**
vehicle 4	1 2 **3 4** 5	1 **2 3** 5	**4**	**1** 2 **3** 4 **5**
vehicle 5	1 2 **3 4** 5	**4**	1 **3** 5	**1** 2 **3** 4 **5**

^1^ in bold active GNSS.

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
