# Peer review of "Scalable Distributed State Estimation in UTM Context"

_sensors, 2020, doi:10.3390/s20092682_

Round 1
Reviewer 1 Report
The presented paper has been devoted to the refinement of the approach to the navigation of the UAS fleet. The key point of the authors' idea is facilitating of processing the vast volume of navigation data, without a Central Processing Unit. The original idea is clusterization that obeys the suggested criterion. According to the proposed solution, only locally relevant computation is required to take place in each local processing unit. The core of the considered method is the well-known Kalman filter, which was adjusted to reduce computational load. The methodology proposed in this article combines the results achieved for scalable decentralized systems obtained by the Internodal Transformation Theory methodology with the advantages guaranteed by the use of consensus-based techniques.
There are some minor corrections and pieces of advice:
1. The designators sUAV, SUAV, UAV should be the same throughout the text.
2. What is the abbreviation EGNSS mean?
3. Row 294 is missing some text.
4. I would not say that the inertial reference system is conventional for UAV movements description (2.6). Yes, by far, it is valid for IMU. However, it is not the case for GNSS, which typically works in geocentric system WGS84. For monitoring or mapping, we use mostly the last one.
5. There are 78 expressions in total in text, but the references for these expressions are quite modest. I am not sure that all those expressions were drawn by authors, so additional references from the references list are highly recommended.
6. Of course, the simulation data are not enough to prove the authors' conclusions entirely, but it seems that it may be a good theme for the publications in what follows.
Reviewer 2 Report
The authors of the manuscript proposed an original algorithm solution for the safety coordination of mutually independent UAV flights. This is a key advantage of the study. It is appropriate to apply the method where several UAVs are flying simultaneously in a limited area. Such cases are possible in the natural and cultural objects visited by tourists or by conducting special environmental space research.
The idea is innovative, and the authors of the manuscript have achieved their goal - to improve the UAV air traffic control methodology.
One question is whether the method used takes into account and allows the identification and avoidance of accidental flying obstacles, such as bird assemblages or individual large birds.
Reviewer 3 Report
The paper is interesting and brings a relative novelty approach for UAS traffic management.
Concerning the paper I a have several small remarks:
1. line 92 I am missing some more citations.
2. Did the author perform any testing in a real situation or still only the design of the algorithm without implementation?
3. I there any limit in the number of UAVs connected (size of the fleet)?
4. How much will your approach influence the battery drain of the UAVs? In other words whats the comparison with the usage of local computational
resources only?
5. What is the price drop between "normal" UAV and yours with desired sensors?
6. Some equations are using variables, which will be better to explain under the equations itself.
7. Ore some of the already available drones equipped with all desired sensors to be capable of used your approach?
